# LCPPO: An Efficient Multi-agent Reinforcement Learning Algorithm on Complex Railway Network

**Primary Keywords:** *(1) Applications; (2) Learning; (7) Multi-Agent Planning*

## Abstract

The complex railway network is a challenging real-world multi-agent system usually involving thousands of agents. Current planning methods heavily depend on expert knowledge to formulate solutions for specific cases and are therefore hardly generalized to new scenarios, on which Multi-agent Reinforcement Learning (MARL) draws significant attention. Despite some successful applications in multi-agent decision-making tasks, MARL is hard to be scaled to a large number of agents. This paper rethinks the curse of agents in the centralized-training-decentralized-execution paradigm and proposes a local-critic approach to address the issue. By combining the local critic with the PPO algorithm, we design a deep MARL algorithm denoted as Local Critic PPO (LCPPO). In experiments, we evaluate the effectiveness of LCPPO on a complex railway network benchmark, Flatland, with various numbers of agents. Noticeably, LCPPO shows prominent generalizability and robustness under the changes of environments.

## Introduction

Multi-agent Reinforcement Learning (MARL) has drawn significant attention in multi-agent decision-making tasks, e.g. continuous control on robots (Yan et al. 2023), playing strategic video games (Wang et al. 2022b), distributed voltage control on grid networks (Wang et al. 2022a) and cooperation in autonomous driving (Keviczky et al. 2007). Although MARL has sparked significant interest in the community, its successful applications are primarily concentrated in cases where the number of agents is limited (less than 10). Most existing MARL algorithms still suffer from increasing complexity with more agents in the system. This can partially explain why MARL is still unable to master the complex railway networks, where there exist at most thousands of agents. Flatland (Mohanty et al. 2020) is an open-source platform simulating traffic on complex railway networks. In this platform, MARL has not yet outperformed the traditional optimization approaches, which motivates us to specifically design a more efficient paradigm assisting MARL to address the real-world problem in this work.

In this paper, we begin by investigating the reason why existing MARL algorithms would fail on complex railway networks. The training of the multi-agent systems is a non-stationary stochastic process from the single agent's perspective so that independent learning (Claus

and Boutilier 1998) will receive an unstable training process. To address this issue, MARL algorithms heavily rely on the Centralized-Training-Decentralized-Execution (CTDE) (Oliehoek, Spaan, and Vlassis 2008) paradigm. Based on CTDE, each agent can gather information from other agents during training (i.e., coordinating and communicating with other agents). This information is encoded in agents' policies so that they can still perform harmoniously with local observations during execution. Figure 1 provides an intuitive example of independent learning and CTDE on actor-critic-based methods. Figure 1b visualizes the independent learning that each agent has an independent critic with its own observation and action as inputs, denoted "independent critic". Figure 1c concludes most popular CTDE-based methods (Lowe et al. 2020; Wang et al. 2020a; Sunehag et al. 2018) in the MARL society. There exists a global mixer gathering all other agents' actions and observations. The global information is then fed into the critic network (denoted "global critic") to produce a more consistent value prediction and eliminate the non-stationarity. Nevertheless, the complexity of the global critic grows along with the number of existing agents, which results in the global information being redundant so that the learning procedure would be unstable in practice (Yu et al. 2022). On the other hand, both paradigms of forming critics do not utilize the physical information existing in the physical system (e.g. the group structure 1a from the railway network). In this work, we propose the local critic 1d paradigm by taking advantage of the provided group structure depicting the spatial relationship among agents to mitigate the above issue incurred by the global critic.

Nevertheless, there exist several challenges to directly applying the local critic to MARL. First, the group structure (e.g. members, connections, size) is non-static and each agent may be involved in different groups throughout the whole process, which prompts the introduction of a novel mixing network to deal with variations of the group structure. The output of this mixing network usually has a practical implication as the group long-term value in previous works (Sunehag et al. 2018; Rashid et al. 2018), which is difficult to track in this scenario when the group is formalized and unravelled temporarily. The second challenge is how to appropriately clarify the contribution of each agent to a certain group. When an agent had joined different groups

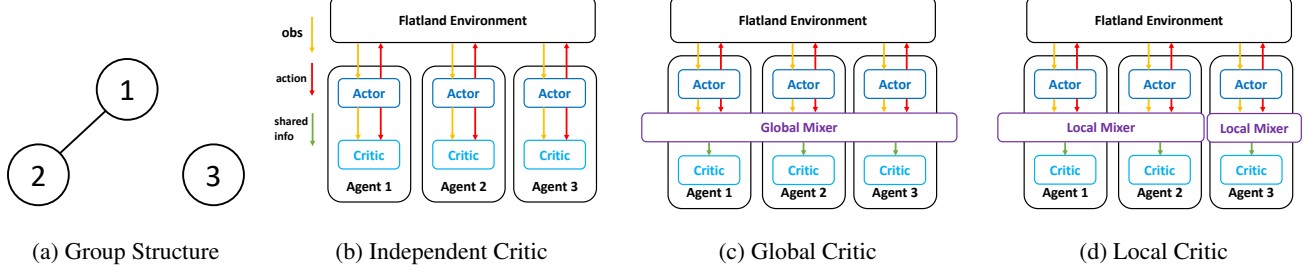

Figure 1: Different Types of Critics in Multi-agent Systems.

along a trajectory, it would be non-trivial to study which past groups or group members its long-term rewards might be influenced by, which directly impacts the policy optimization.

Our main contribution is to sweep away the practical issues of incorporating the local critic into multi-agent systems, namely dynamic group and agent coordination. Our method is easy to implement on actor-critic frameworks like PPO, leading to a practical MARL algorithm LCPPO. Furthermore, our approach demonstrates superior performance over other MARL baselines on a complex railway network simulator Flatland with various numbers of agents. Finally, our method shows additional generalizability and robustness regarding environmental changes in the network system, which reveals LCPPO to be a promising approach in real-life applications.

## Related Work

Vehicle Planning Problem as modelled in the Flatland Environment has been an active research area within the operations research (OR) community dating back to decades (Bodin and Golden 1981; Ryan and Foster 1981). To the Flatland challenge, the winning solution in 2020 (Laurent et al. 2021) was from the perspective of Multi-Agent Path Finding (MAPF) (Stern et al. 2019) combining it with other optimisation techniques. For example, to handle malfunctions, an improved version of Minimum Communication Policies (MCP) (Ma, Kumar, and Koenig 2016) was used to avoid the deadlocks by stopping some trains to maintain the order that each train visits each location. Overall, most mentioned OR methods heavily depend on expert knowledge to formulate solutions for specific cases and are therefore hardly generalized to new scenarios. Moreover, they all need global information for planning, which is inefficient and unstable for unforeseen situations.

The ever-growing complexity of railway networks and a need for real-time rescheduling makes OR methods infeasible and has paved the way for MARL solutions owing to their success in optimisation problems. However, scalability to a large number of agents and efficient coordination of individual agents remain major challenges. Currently, two parallel approaches are attempting to manage both challenges. The first approach is learning decentralized policies and adding communication between agents (Foerster et al. 2016; Sukhbaatar, Szlam, and Fergus 2016; Das et al. 2019). These methods are sometimes bothered by the communication bandwidth and latency and are not appropriate for railway systems where agents are far away. The other approach is centralized-training-decentralized-execution (CTDE) introduced in Background. To implicitly handle coordination during training, MARL methods usually require a global critic to gather all agents' information (Lowe et al. 2020), or decompose a global critic into individual value functions (Sunehag et al. 2018; Rashid et al. 2018, 2020; Wang et al. 2020b). All of them suffer from large joint state-action space and cannot scale to the number of agents. This directly motivates the local critic approach proposed in this paper. Yang et al. (2018) have already investigated issues of large scalability in MARL, unfortunately, which simplifies agents based on static neighbouring information and is difficult to apply on dynamic railway network setups.

## Background

### Multi-Agent Reinforcement Learning

Multi-agent reinforcement learning (MARL) is a domain that combines multi-agent learning and reinforcement learning to solve a game model depicting a realistic problem. In this work, we apply MARL as a basic learning framework to solve the complex railway network. Following the common setting in MARL, we model the multi-agent system (MAS) as a partially observable stochastic game (POSG) which can be expressed as the following 7-tuple (Kumar and Zilberstein 2009) such that $\langle \mathcal{N}, \mathcal{S}, \mathcal{A}, \mathcal{O}, \{r_i\}_{i \in \mathcal{N}}, T, b_0 \rangle$. More specifically, $\mathcal{N} = \{1, 2, ...\}$ is a set of agents existing in the MAS. $\mathcal{S}$ is a set of available states. $\mathcal{O} = \times_{i \in \mathcal{N}} \mathcal{O}_i$ is a joint observation set, where $\mathcal{O}_i$ is agent $i$'s observation set; while $\mathcal{A} = \times_{i \in \mathcal{N}} \mathcal{A}_i$ is a joint action set, where $\mathcal{A}_i$ is agent $i$'s action set. Each agent $i$ is equipped with a reward function to evaluate its performance such that $r_i : \mathcal{S} \times \mathcal{A} \to \mathbb{R}$. Additionally, the transition function of the MAS can be described as follows: $T : \mathcal{S} \times \mathcal{A} \to \Delta(\mathcal{S} \times \mathcal{O})$, where $\Delta(\mathcal{X})$ is the set of all probability distributions defined over a set $\mathcal{X}$. $b_0 \in \Delta(\mathcal{S})$ is the initial state distribution. The objective of POSG is to maximize each agent's individual discounted cumulative rewards by a stationary policy $\pi_i : \mathcal{O}_i \to \mathcal{A}_i$ such that $\max_{\pi_i} \mathbb{E}[\sum_{t=0}^{\infty} \gamma^t r_i(s_t, a_t)]$, where $\gamma \in (0, 1)$ is a discount factor, $s_t \in \mathcal{S}$ and $a_t \in \mathcal{A}$. In MARL, the usual learning paradigm to solve POSG is called the multi-agent actor-critic framework, for which each agent individually applies the actor-critic framework to optimize its policy. Two of the most popular algorithms based on this paradigm are IPPO

(de Witt et al. 2020) and MAPPO (Yu et al. 2022), which extends the vanilla multi-agent actor-critic framework by incorporating the PPO algorithm (Schulman et al. 2017). In this work, we propose Local Critic PPO based on MAPPO via formalizing the critic with GNNs to capture sufficient information from the complex railway network.

## Centralized Training Decentralized Execution

MARL algorithms are applied either as fully centralised methods where a single policy with joint action is learned for all agents or in an independent agent learning setting - also called decentralised learning where agents are optimised separately. Nevertheless, the fully centralised method could lead to the curse of dimensionality to impede learning the optimal joint policy, while the independent learning (e.g. IPPO) may result in the non-stationary learning procedure (Hernandez-Leal, Kartal, and Taylor 2019). To trade off the benefits and drawbacks of these two paradigms, Centralised Training and Decentralized Execution (CTDE) (Oliehoek, Spaan, and Vlassis 2008) (e.g. MAPPO) was proposed to form each agent's critic by all other agents' information (e.g. observations and actions), still maintaining the decentralised policies to approximate the joint policy as used in independent learning paradigm to avoid the curse of dimensionality. Standing by the view of application, a limitation of CTDE is that it always collects the information of all agents to form a critic for an agent $i$, however, in physical scenarios some agents could not influence agent $i$. This would inevitably cause some unnecessary fluctuations on the approximate critic, leading to potential learning instability (Yu et al. 2022). To mitigate this issue, we propose the local critic to aggregate the *sufficient* agents' information, based on the existing physical information (e.g. a tree structure describing the spatial relationship among agents) provided by the complex railway network. This would directly filter out the information of irrelevant agents, to reduce the instability induced especially from the scenarios with a large number of agents. The outstanding performance of the proposed local critic sheds light on the necessity of incorporating known physical information into design when dealing with real-world problems.

# Local Critic Multi-agent Reinforcement Learning (LCMARL)

Introduced in Background, MAPPO depends on a global critic during training, which fails to scale on complex railway networks like Flatland with more than 10 agents. In this section, we take an alternative perspective, which formalizes a local group and constructs a local critic for training. Incorporating the local critic method into the popular RL algorithm PPO (Schulman et al. 2017), we achieve a practical MARL algorithm applicable to the large-scale railway planning problems, denoted as **L**ocal **C**ritic **PPO** (LCPPO).

## Overview

Figure 2 provides an overview of the overall approach of incorporating the local critic into the MARL framework, in particular actor-critic-styled algorithm.

Suppose $N$ agents in the system, and they receive their local observations $\boldsymbol{o_t} = (o_t^1, o_t^2, \ldots, o_t^N)$ at each step $t$. Without loss of generality, we assume global state $s_t = \boldsymbol{o_t}$. However, from the single agent's view, the system is still partially observable. The group structure $g_t$ is a graph representation with $N$ nodes and $E$ edges, which can be naturally constructed in the railway system. In specific, two agents share an edge if they are on rails connected by less than one crossroad. For each agent $i$, the number of its neighbouring agents $\mathcal{N}_t(i)$ is usually much less than $N$ due to the sparsity property in railway systems.

All observations are further passed into the local-critic network $V(\boldsymbol{o_t}, g_t; \phi) : \mathcal{O} \times \mathcal{G} \to \mathbb{R}$ to predict agents' individual values $\boldsymbol{v_t} = (v_t^1, v_t^2, \ldots, v_t^N)$, where $\mathcal{G}$ is the space of group structure. The local-critic network is represented as a neural network parameterized with $\phi$, as illustrated in Figure 2b. The network utilizes the group structure $g_t$ within the GNN (Scarselli et al. 2009) layer. For each agent $i$, it ensures its local information can only flow inside its neighbouring agents $\mathcal{N}_t(i)$. If the neighbouring size is limited to $G$ and the number of agents $N$, the complexity of GNN is $\mathcal{O}(GN)$ compared with $\mathcal{O}(N^2)$ of the global critic in Figure 1c, and thus easily speedup on hardware (Wang et al. 2023) when $G \ll N$. In practice, the GNN structure is implemented with the transformer (Vaswani et al. 2017) architecture with the mask mechanism.

## Dynamic Group

The biggest challenge in learning the local-critic network is the evolving group structure $g_t$. For agent $i$, it's urgent to discover its influence on its neighbouring agents $\mathcal{N}_t(i)$ in several successive steps, but $\mathcal{N}_t(i)$ can change at every step. To mitigate this issue, we propose a concept of the imaginary step $\tilde{t}$ (red dashed frame in Figure 2a). The imaginary step $\tilde{t}$ utilizes the observations $\boldsymbol{o_{t+1}}$ but maintains the group structure $g_t$. By passing through the local-critic network, we obtain virtual values $\tilde{\boldsymbol{v}}_{t+1} = V(\boldsymbol{o_{t+1}}, g_t)$. Since values $\tilde{\boldsymbol{v}}_{t+1}$ and values $\boldsymbol{v_t}$ (individual values at step $t$) are calculated with the same group structure $g_t$, there exists an iterative relation with these values and rewards $\boldsymbol{r_t}$, which will be further introduced in the next section.

With the introduction of the imaginary step, values at different steps are connected with the same group structure. In detail, assume the predicted values at step $t$ are $\boldsymbol{v_t} = V(\boldsymbol{o_t}, g_t; \phi)$ for all agents, while the imaginary values at step $t+1$ are $\tilde{\boldsymbol{v}}_{t+1} = V(\boldsymbol{o_{t+1}}, g_t; \phi)$. These values represent the same meaning: the discounted expected cumulative return agents can achieve under the static group structure $g_t$. We denote the $i$-th output of $V(\boldsymbol{o_t}, g_t; \phi)$ as $V_i(\boldsymbol{o_t}, g_t; \phi)$ for simplicity. According to the dynamic programming techniques (Sutton and Barto 2018), the extended Bellman equation for any $i$ on value function $V_i$ can be derived:

$$V_i(\boldsymbol{o_t}, g_t; \phi) = \mathbb{E}_{\boldsymbol{a_t}, \boldsymbol{r_t}, \boldsymbol{o_{t+1}}}\big[r_t^i + \gamma(V_i(\boldsymbol{o_{t+1}}, g_t; \phi)\big], \quad (1)$$

where $\gamma$ is the discount factor to account for future steps. The expectation is concerning the next observations, actions and rewards. The complicated expectation computation is

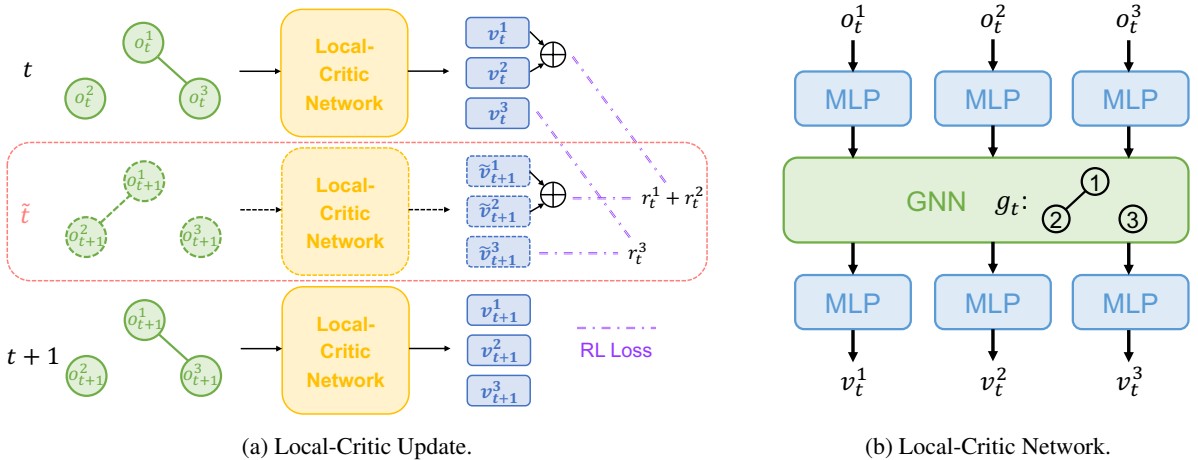

(a) Local-Critic Update.     (b) Local-Critic Network.

Figure 2: An overall of Local-Critic Multi-agent Reinforcement Learning (LCMARL).

usually approximated by sample-based methods (Sutton and Barto 2018).

## Agent Coordination

Equation 1 indicates that the value of agent $i$ is only accounted for by the received individual reward $r_t^i$. It is imperfect since agents could reach a local sub-optimal solution as Prisoner's dilemma in Game Theory. Instead, we would like to encourage agents to find solutions better for global interest.

Inspired by the VDN method by Sunehag et al. (2018), they utilized the monotonicity of the addition calculation and summed all individual values $V_i$ as the global value, to encourage cooperative behaviours among agents. Similarly, we sum all individual values inside each agent $i$'s neighbourhood to encourage agent coordination inside this local group, which requires much less computation compared with VDN on a global group. Moreover, with the evolution of the local group, agents will possibly have a chance to coordinate with different agents when necessary. The modified Bellman equation takes place on the local-group level instead of on the single-agent level:

$$
\sum_{j \in \mathcal{N}_t(i)} V_j(\boldsymbol{o_t}, g_t; \phi) = \mathbb{E}_{\boldsymbol{a_t}, \boldsymbol{r_t}, \boldsymbol{o_{t+1}}} \Big[ \sum_{j \in \mathcal{N}_t(i)} r_t^j
$$
$$
+ \gamma \sum_{j \in \mathcal{N}_t(i)} V_j(\boldsymbol{o_{t+1}}, g_t; \phi) \Big]. \tag{2}
$$

Intuitively, agent $j$ learns to maximize its own value $V_j$, which also contributes to the local group value.

## Practical Algorithm: LCPPO

Equation 2 describes the recursive definition (Bellman Equation) of the value function with a local-critic perspective, which can be further used to in policy evaluation. For a complete RL algorithm, policy improvement is also required to learn a better policy. In this paper, we rely on the successful single-agent RL algorithm PPO (Schulman

---

**Algorithm 1: LCPPO**

1: **Input:** initial parameters $\theta_0$ for policy function $\pi$, initial parameters $\phi_0$ for value function $V$
2: **for** $k = 1, 2, \cdots, K$ **do**
3:  Set data buffer $\mathcal{D}_k = \emptyset$
4:  **for** $j = 1, 2, \cdots, J$ **do**
5:    Collect trajectory $\tau_j = \{\boldsymbol{o_0}, \boldsymbol{a_0}, \boldsymbol{r_0}, g_0, \boldsymbol{o_1}, \cdots, \}$ by executing actions $\boldsymbol{a_t} \sim \pi(\boldsymbol{a_t}|\boldsymbol{o_t}; \theta) = \prod_{i=1}^{N} \pi(a_t^i|o_t^i; \theta)$ in the environment at each step $t$
6:    **for** each step $t$ and each agent $j$'s neighbouring group $\mathcal{N}_t(j)$ derived from $g_t$ **do**
7:      Compute values $v_t^{\mathcal{N}_t(j)} = \sum_{i \in \mathcal{N}_t(j)} V_i(\boldsymbol{o_t}, g_t; \phi)$
8:      Compute virtual values $\tilde{v}_{t+1}^{\mathcal{N}_t(j)} = \sum_{i \in \mathcal{N}_t(j)} V_i(\boldsymbol{o_{t+1}}, g_t; \phi)$
9:      Compute local group reward $r_t^{\mathcal{N}_t(j)} = \sum_{i \in \mathcal{N}_t(j)} r_t^i + \gamma(\tilde{v}_{t+1}^{\mathcal{N}_t(j)} - v_{t+1}^{\mathcal{N}_{t+1}(j)})$
10:     Compute advantage estimates $\hat{A}_t^{\mathcal{N}_t(j)}$ via GAE (Schulman et al. 2017) with local group reward $r_t^{\mathcal{N}_t(j)}$ and value $v_t^{\mathcal{N}_t(j)}$
11:     Compute rewards-to-go $\hat{R}_t^{\mathcal{N}_t(j)} = \hat{A}_t^{\mathcal{N}_t(j)} + v_t^{\mathcal{N}_t(j)}$
12:     $\tau_j \leftarrow \tau_j \cup \{v_t^{\mathcal{N}_t(j)}, \hat{A}_t^{\mathcal{N}_t(j)}, \hat{R}_t^{\mathcal{N}_t(j)}\}$
13:    **end for**
14:   $\mathcal{D}_k \leftarrow \mathcal{D}_k \cup \{\tau_j\}$
15:  **end for**
16:  Update value function's parameters $\phi$ with Adam optimizer (Kingma and Ba 2015) by fitting rewards-to-go:

$$
\phi_{k+1} = \arg\min_{\phi} \sum_{\tau \in \mathcal{D}_k} \sum_{t=0} \sum_{\mathcal{N}_t(j)} \Big(\sum_{i \in \mathcal{N}_t(j)} V_i(\boldsymbol{o_t}, g_t; \phi) - \hat{R}_t^{\mathcal{N}_t(j)}\Big)^2
$$

17:  Update policy function's parameters $\theta$ with Adam optimizer (Kingma and Ba 2015) by maximizing multi-agent PPO objective:

$$
\theta_{k+1} = \arg\max_{\theta} \sum_{\tau \in \mathcal{D}_k} \sum_{t=0} \sum_{\mathcal{N}_t(j)} \sum_{i \in \mathcal{N}_t(j)} \Big(c_t^i(\theta)\hat{A}_t^{\mathcal{N}_t(j)}, \mathbf{clip}(r_t^i(\theta), 1-\epsilon, 1+\epsilon)\hat{A}_t^{\mathcal{N}_t(j)}\Big)
$$

    where $c_t^i(\theta) = \frac{\pi(a_t^i|o_t^i; \theta)}{\pi(a_t^i|o_t^i; \theta_k)}$

et al. 2017) as the backbone, and develop a novel MARL algorithm, denoted as **L**ocal-**C**ritic **PPO** (**LCPPO**). The specific procedure is explained in Algorithm 1. We assume homogeneous agents and the policy function is $\pi_i(o_t^i; \theta) = \pi(o_t^i; \theta)$ for any agent $i$ with parameters $\theta$ and value function is $V(\boldsymbol{o_t}, g_t; \phi)$ with parameters $\phi$. Agent $i$'s individual value function is denoted as $V_i(\boldsymbol{o_t}, g_t; \phi)$, which is the $i$-th output of $V(\boldsymbol{o_t}, g; \phi)$. LCPPO can be extended to heterogeneous agents in future work with individual policy and value functions. For each agent $j$'s neighbouring group $\mathcal{N}_t(j)$ derived from group structure $g_t$, the group value is defined as the sum of group members' individual values: $v_t^{\mathcal{N}_t(j)} = \sum_{i \in \mathcal{N}_t(j)} V_i(\boldsymbol{o_t}, g_t; \phi)$. The key modification to PPO method is on Line 9 that the local group reward $r_t^{\mathcal{N}_t(j)}$ is modified with an additional correction term $\gamma(\tilde{v}_{t+1}^{\mathcal{N}_t(j)} - v_{t+1}^{\mathcal{N}_{t+1}(j)})$. This term is designed to compensate the calculations on advantages $\hat{A}_t^{\mathcal{N}_t(j)}$ so that it accounts for virtual values instead of real values to follow Equation 2.

# Experiments

## Experimental Setup

**Task Description** We evaluate the LCPPO on Flatland (Mohanty et al. 2020), a simplified grid environment to simulate the railway networks with an easy-to-use machine learning interface. The goal is to control each vehicle with different routes to arrive safely and punctually. Figure 4 visualizes the running process in Flatland. We mainly follow the official environmental configurations [1] with 10/20/30 agents respectively. In particular, the map size is $30 \times 30$ with 3 cities (2 cities for 10 agents). The max rails between cities are 2 and there are 2 rail pairs in each city. The malfunction rate is 0 and the speed for the vehicle is 1.0 grid per step, and verified in later analysis.

Regarding the MARL setup, we follow the previous setup (Jiang et al. 2022) that each agent $i$ receives a local observation $o_t^i$ at step $t$ consisting of two parts: agent attributes $X^{\text{attr}}$ and tree-structured representation $X^{\text{tree}}$. $X^{\text{attr}}$ describes the individual attributes of each agent with 83 dimensions, e.g. scheduled departure and arrival time. $X^{\text{tree}}$ represents the spatial information on the grid environment, which is encoded as the tree structure $X^{\text{tree}} = (\text{node}_{v=1}^V, \text{edge}_{e=1}^E)$ includes $V = 31$ nodes with 12-dimensional node attributes $\text{node}_v$ and $E = 30$ edges with 3-dimensional attributes $\text{edge}_e$ indicating connected nodes. All the information is derived from the spanning tree, which is constructed by traversing from the agent's location and branch at each possible crossroad. Please refer to Jiang et al. (2022) for the detailed description of the spanning tree and attributes. The action space includes five discrete actions: do nothing, go forward, stop, turn left, and turn right. Regarding the group structure needed by LCPPO during training, it's defined as follows: for each agent, any other agents who appear in the first level of its spanning tree belong to the same group. The common group size is less than 5, which is much less than the total number and guarantees the efficiency of LCPPO.

[1] https://flatland.aicrowd.com/challenges/neurips2020/envconfig.html

**Evaluation Metric** We adopt multiple objectives to evaluate the performance of different methods. Each agent receives an individual reward signal at each step, consisting of the following items:

- **Arrival Reward:** $r_t^a = 1$ if the agent reaches the target and $r_t^a = 0$ otherwise;
- **Deadlock Penalty:** $r_t^l = -1$ if the agent immerses in a deadlock and $r_t^l = 0$ otherwise. A deadlock happens when two trains step into a single trail from opposite directions. The deadlock quickly blocks the rails and catastrophically paralyzes the whole system, and thus should be penalized.
- **Environment Reward:** To encourage the train to arrive on time, Flatland environment (Mohanty et al. 2020) provides an environmental reward defined as

$$r_t^e = \begin{cases} 1.0, & \text{if } t \leq B \text{ AND new arrival} \\ (B-t)/T_{\max} + 1, & \text{if } B < t < T_{\max} \text{ AND new arrival} \\ (B-d)/T_{\max}, & \text{if } t = T_{\max} \text{ AND not arrival} \\ 0, & \text{otherwise} \end{cases} \quad (3)$$

where $B$ is the latest arrival time, $T_{\max}$ is the system's maximum running steps and $d$ is the shortest path Manhattan distance between the train's position and its target at $T_{\max}$. The intuition of the reward is to punish the delays after the scheduled latest time.

The final reward for agent $i$ is the weighted sum of all terms above: $r_t^i = c_e r_t^e + c_a r_t^a + c_l r_l^t$, where $c_e = 1.0$, $c_a = 5.0$ and $c_l = 2.5$ follows previous work (Jiang et al. 2022).

## Baselines and Implementation

- **IPPO** implements the structure as in Figure 1b. Each critic only relies on local observation during training.
- **MAPPO** represents the structure as in Figure 1c and previous work (Yu et al. 2022). The critic network gathers all agents' observations as input and predicts the value.
- **LCPPO** follows Algorithm 1 introduced in this paper. Theoretically, the critic network only utilizes observations from its neighbours. Practically, we use all agents' observations and adopt a Transformer (Vaswani et al. 2017) layer with the mask mechanism to imitate the effects.

Table 1: Hyperparameters of all baselines.

| HYPERPARAMETERS | VALUE |
|---|---|
| BATCH SIZE | 1000 |
| GAE LAMBDA | 1.0 |
| KL DIVERGENCE COEFFICIENT | 0.2 |
| KL TARGET VALUE | 0.1 |
| VALUE FUNCTION COEFFICIENT | 1.0 |
| VALUE FUNCTION CLIP | 10.0 |
| NUMBER OF GRADIENT ITERATION | 10 |
| GRADIENT NORM CLIP | 0.1 |
| LEARNING RATE | 5E-4 |
| TRAINING STEPS | 6E6 |

For a fair comparison, all baselines share the same actor network structure of a 2-layer feedforward neural network

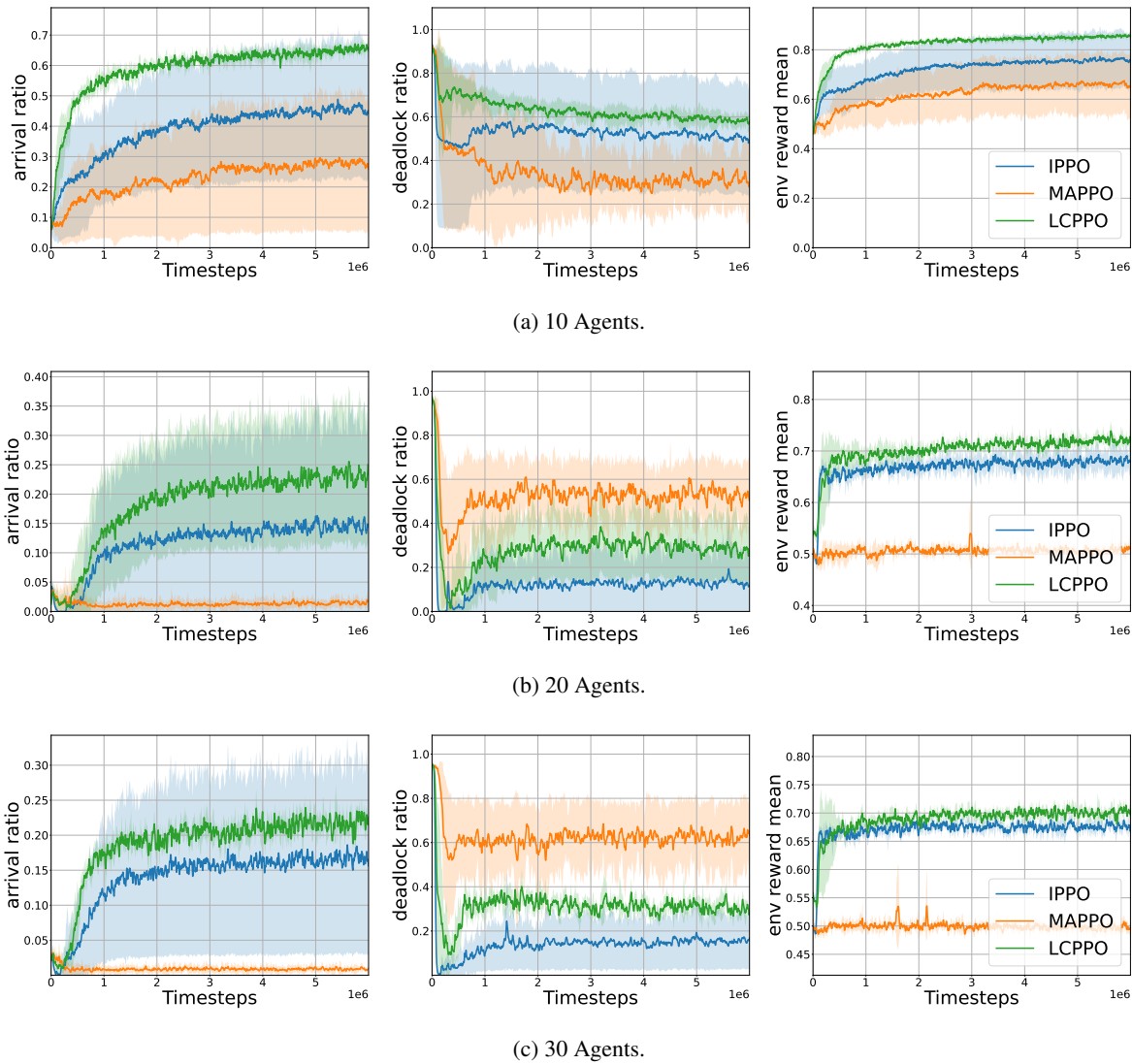

(a) 10 Agents.

(b) 20 Agents.

(c) 30 Agents.

Figure 3: The training performances of all baselines on the Flatland simulator with variant numbers of agents. All experiments are carried out with 5 random seeds and the average performances across all agents are plotted with standard deviation as shaded area.

with 64 hidden units each. Notably, the parameter sharing technique (de Witt et al. 2020) is enabled among agents for efficient learning. The critic network has a hidden-layer structure as the actor network, and there is an additional transformer layer to group local information in LCPPO. Other hyperparameters are demonstrated in Table 1.

## Main Results

The main results of 10/20/30 agents are shown in Figure 3. All experiments are carried out with 5 random seeds and the average performances are plotted with standard deviation as shaded area. All evaluation metrics are averaged across participating agents. In terms of the arrival ratio and environmental reward, all experiments share a similar trend of LCPPO > IPPO > MAPPO, which is strong evidence that

the local critic successfully guides the coordination of agents thus leading to more on-time arrivals. Notably, MAPPO can't learn anything with the increasing number of agents. This result complies with the curse of agent issues occurring in the MARL area as explained in the Introduction.

## Generalization

Generalization (Kirk et al. 2023) is essential for learning-based methods since there might be mismatches between training and testing environments in practice, which also applies to the railway system. There are various malfunctions in real-world railway trails. Besides, it's common to add or reduce train routes, which all require rescheduling plans. In theory, LCPPO only utilizes local information to guide planning behaviours. When mismatches happen in the system,

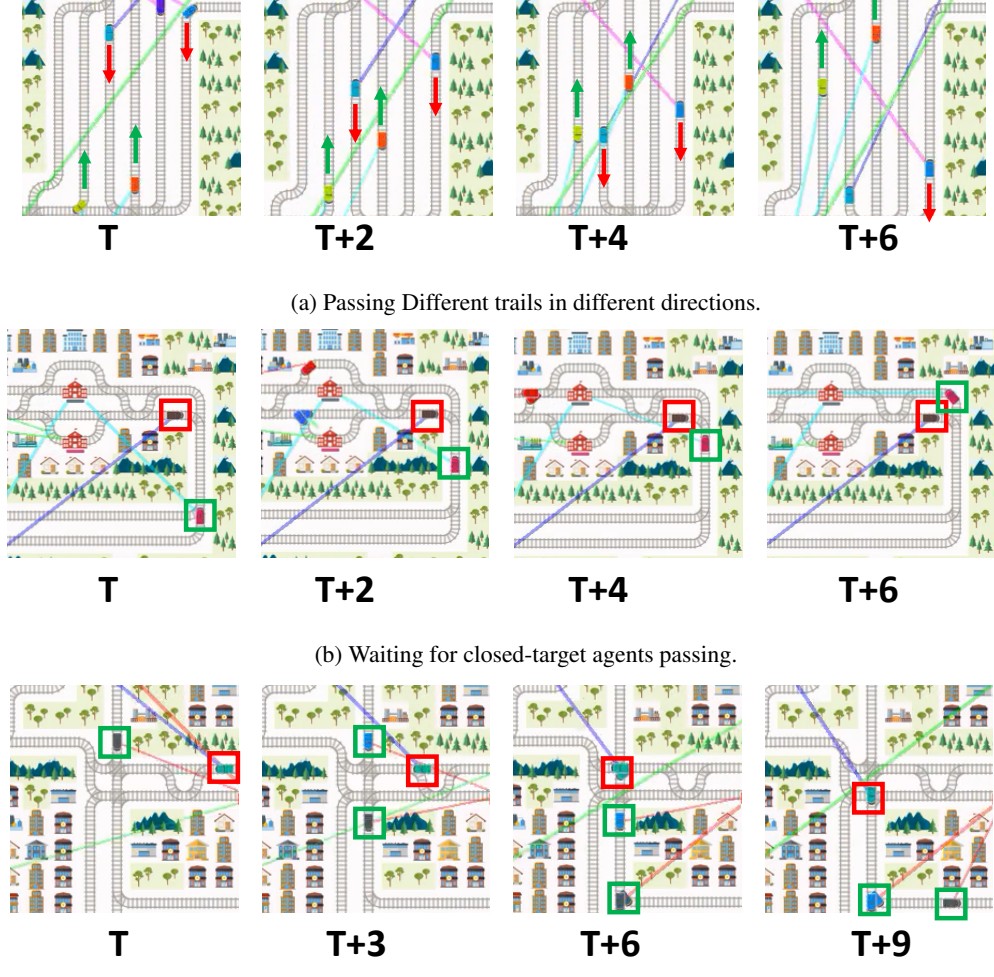

(a) Passing Different trails in different directions.

(b) Waiting for closed-target agents passing.

(c) Waiting for same-direction group agents passing.

Figure 4: Visualization of agents' policies learned by LCPPO algorithm.

local groups close to the mismatches need re-planning and other groups are not influenced. Compared with the global critic method, a malfunction could influence all agents since it hasn't been during training.

To demonstrate the generalization of LCPPO, we design the following experimental setups: we first utilize different algorithms to train planning policies on environments defined in the setups. Later on, certain components of the environment are modified to simulate the mismatch in the system and all policies are tested on newly changed environments without further tuning. Regarding the changing components, we consider the following scenarios:

- **Malfunctions:** Trains are randomly stopped for random duration. The stopped train would block the trail and block other trains passing. This stochastic process follows the Poisson process. The mean rate of the Poisson process is 0.0001. The stopping duration ranges from 15 steps to 50 steps.
- **Speeds:** All trains have speed with one grid per step dur-

ing training. During testing, 1/4 of trains maintain this speed, while 1/4 with one grid per 2 steps, 1/4 with one grid per 3 steps and 1/4 with one grid per 4 steps.

- **Agents:** There are 20 trains in the network during training. 10 more agents are added during testing to challenge the generalization ability.

Table 2: The average arrival ratio of all baselines under different test scenarios.

| Algorithms | Test Scenarios | | |
|---|---|---|---|
| | *Malfunctions* | *Speeds* | *Agents* |
| **IPPO** | 0.160 ± 0.188 | 0.124 ± 0.153 | 0.110 ± 0.135 |
| **MAPPO** | 0.016 ± 0.006 | 0.033 ± 0.008 | 0.016 ± 0.005 |
| **LCPPO** | 0.235 ± 0.113 | 0.194 ± 0.100 | 0.181 ± 0.092 |

All experiments are carried out with 5 random seeds and we report the average arrival ratio and its standard deviation in Table 2. Apparently, LCPPO is the most robust

algorithm among all baselines. The global critic method (MAPPO) is the least favourite method under environmental mismatches. This proves our concerns about current state-of-the-art MARL methods. The number of agents is the most influential factor to all baselines, which calls theories from open team research (Rahman et al. 2021).

## Conclusion

This paper focuses on the applications of MARL on complex network railway networks. The failure of state-of-the-art MARL methods in such a large-scale environment directly motivates this work. We proposed the local critic idea and achieved an efficient MARL algorithm LCPPO. LCPPO scales efficiently with the number of agents on the Flatland challenge and performs better and more robustly than other baselines.

Despite the advantages provided by the local critic, LCPPO still renders some deadlocks and unsuccessful plannings, which is non-negligible in real-world applications. It implies that the CTDE paradigm might not be enough to handle the coordination on agents (Zhou et al. 2023). Therefore, it would be beneficial to include communications among local groups or global information (graph structure, other agents' observations...) during execution for global optimal solutions. Besides, current updates on the value function rely on the sum of rewards in the local group, which treats all agents with identical importance. This assumption might be wrong for heterogeneous multi-agent systems. A more advanced credit assignment technique should be considered (Rashid et al. 2018; Wang et al. 2022b) and extended to dynamic group scenarios.

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
