# OpenReview forum: "LCPPO: An Efficient Multi-agent Reinforcement Learning Algorithm on Complex Railway Network"
_icaps-conference.org/ICAPS/2024/Conference — ICAPS 2024_

### Official Review · Reviewer_wb37 · 2024-01-12

**Significance And Importance:** 2
**Soundness:** 3
**Novelty:** 3
**Clarity:** 3
**Overall Evaluation:** 1
**Confidence:** 3

**Weaknesses:**

1: Minor weaknesses that are easily fixable.

**Contributions Of The Paper:**

Two contributions intertwine in this paper.  On the one hand, the authors present an extension of PPO to have local critics.  This helps to scale the algorithm to large distributed agent scenarios.  On the other hand, the authors apply this LCPPO algorithm to scheduling in the Flatland domain.

The paper's title is ambiguous whether the authors think the contribution is a new algorithm, LCPPO -- in which case the algorithm should be tested on more than one domain -- or whether the contribution is a more effective RL approach to Flatland.

The authors argue that the lack of generalisation of engineered meta-heuristics, and their need (usually) for global information make "OR methods" to be "infeasible" for Flatland-like problems.  We could add that such OR methods are more effective on known deterministic problems than when there is stochastics and dynamics such as rescheduling.

The authors however might revisit their claim "The ever-growing complexity of railway networks and a need for real-time rescheduling makes OR methods infeasible and has paved the way for MARL solutions owing to their success in optimisation problems. However, scalability to a large number of agents and efficient coordination of individual agents remain major challenges."  This is because, first, railway networks are not ever-growing in complexity: if nothing else, the physical space on earth is limited!  Second, OR methods remain quite effective, as the winners of previous Flatland Challenges give evidence.  Third, as the authors then say, MARL methods are not without scaling challenges themselves.  Claims such as the above can be expressed less harshly against "OR methods" while still motivating the rationale to study MARL for Flatland.

The authors could comment usefully on the link between expert knowledge (embedded in OR methods) and "the necessity of incorporating known physical information into [MARL] design when dealing with real-world problems."

The above comments said, this reviewer is satisfied by the motivation for LCPPO is a MARL training method.

The exposition seems adequate, and this reviewer opines that the figures are helpful.

The experimental results seem to be derived adequately.  A comparison with non-RL methods would add to the paper.

It is not stated where the source code of LCPPO is available.


Here are some additional comments --

page 2

"Vehicle" --> "The Vehicle"

page 3

MARL "is a domain" <-- why is MARL a "domain" and not a "methodology"?

page 4

remove "as Prisoner’s dilemma in Game Theory"

unwanted space before "Sunehag"

"they utilized the monotonicity" --> "who utilized the monotonicity"

page 5

unwanted space before footnotemark 1

missing space after "Figure 1c"

page 7

"hasn't" --> "has not"



**after author response**

Thanks for listening to the reviewers and responding.  I would be happy to see that table in the paper!

**Ethical Considerations:**

(1) Not Applicable: The paper does not have any ethical considerations to address

**Nomination For Best Paper:**

No

**Questions For Authors:**

Questions are in above text box

**Reproducibility:**

2: Some details are missing, but the paper still appears to be replicable with some effort.

**Strengths Of The Paper:**

Strengths are in above text box

**Weaknesses Of The Paper:**

Weaknesses are in above text box

---

> ### Author Rebuttal · Authors · 2024-01-27
>
> Thank you for your insightful and helpful comments on our paper. Your professional advice on the OR method is extremely helpful in improving our paper. We will rephrase this point and fix typos in the revised version. We hope the following answers can address your other questions and concerns.
>
> A1 (OR and MARL): We fully agree with your views on OR and MARL fields. These two fields are not competitive but highly complementary. We also agree that the comparison against other OR methods is beneficial to the paper! Thus we compare LCPPO with the winning solution of the flatland challenge, thanks to the open-sourced solution: `github.com/Jiaoyang-Li/Flatland`. Notably, we couldn't find detailed implementations of other RL top solutions on this benchmark, so we only baselined standard RL algorithms like IPPO/MAPPO. This makes our work valuable for providing a competitive and easy-to-use RL implementation on this benchmark. We print the average environment reward and time cost for OR and LCPPO methods in the table. The time cost for OR is calculated as the sum of the initial plan and interval replans. The time cost for MARL is calculated as the episode inference time.
>
> | Env Reward / Time Cost (ms) | 10 Agents | 20 Agents  | 30 Agents  |
> | --------------------------- | --------- | ---------- | ---------- |
> | OR                          | 0.93 / 75 | 0.91 / 421 | 0.88 / 862 |
> | LCPPO                       | 0.82 / 75 | 0.71 / 218 | 0.70 / 269 |
>
> Regarding the planning performance, OR consistently outperforms LCPPO, which indicates the big gap left for RL methods. However it is promising to see the time cost for LCPPO doesn't scale as much as the OR method, which results from the CTDE framework. We will add a more detailed comparison in the revised version. The main source code for LCPPO can be found in: `flatland-marllib/marllib/marl/models/zoo/mlp/lc_mlp.py`, where we bring a transformer layer with the mask attention to gather local information.
>
> We also thank your advice on the title, which will be fixed to focus more on the effectiveness on the Flatland environment. We will fix other additional comments in the revised version. Thank you for your time.

---

### Official Review · Reviewer_xVU4 · 2024-01-16

**Significance And Importance:** 1
**Soundness:** 2
**Novelty:** 1
**Clarity:** 2
**Confidence:** 3

**Weaknesses:**

0: Minor weaknesses requiring some work to be addressed for the paper to be accepted.

**Contributions Of The Paper:**

The paper presents a deep multi-agent reinforcement learning (MARL) algorithm that integrates a local critic approach with the proximal policy optimization (PPO) algorithm. The authors state that it is designed to handle large-scale multi-agent systems. They use complex railway networks for demonstration.

Their approach (LCPPO) is evaluated on the Flatland benchmark, a complex railway network simulator. According to the authors, the results demonstrate its effectiveness, generalisability and robustness in different scenarios with different numbers of agents.

The authors compare the performance of LCPPO with a MARL baseline algorithms, showing better performance in terms of generalisability and robustness to environmental changes.

**Ethical Considerations:**

(1) Not Applicable: The paper does not have any ethical considerations to address

**Nomination For Best Paper:**

No

**Overall Evaluation:**

-2: (reject)

**Questions For Authors:**

1. Please define the role of the GNN in your work more clearly.
2. Why did you define the neighboring group as you did? Did you evaluate other generation rules e.g. more than one crossroad.
3. To what extent would you agree that your approach is only an incremental development of MAPPO?
4. With regard to generalization: To what extent are the implemented changes proof of generalization? The change in speed does not seem to play any role at all for generalization or learning, and the malfunctions also address a temporal aspect, which does not come into play in your state description, since an action evaluation takes place in the observation of the agent. Surely no assumptions are made about the future here? There is neither stacking nor any recurrent information flows.

**Reproducibility:**

1: Difficult to reproduce because of missing detail.

**Strengths Of The Paper:**

LCPPO performs better in the defined Flatland benchmark scenarios as classic MARL approaches.

The approach addresses a central problem of large scale multi-agent systems, namely scalability.

**Weaknesses Of The Paper:**

The paper does not allow sufficient reproducibility in its current form. Insufficient information is provided on essential components, e.g. the GNN.

Essentially, the novelty of the paper lies in the fact that not all agents are included, but only a so-called neighboring group. The authors are only partially successful in defining this neighboring group, and there is also a lack of in-depth analysis as to whether other generation rules are more suitable for creating the neighboring group.

The state of the art is insufficiently elaborated. It is very narrowly defined, which is why other works that deal with local vs. global optimization are not taken into account. Further evaluations should have been carried out at this point.

Some of the statements, especially in the introduction, are without sources or a comprehensible explanation. The selection of sources (e.g. strategic video games) also seems arbitrary and does not take up the fundamental work in these areas.

---

> ### Author Rebuttal · Authors · 2024-01-27
>
> Thank you for the helpful reviews on our paper. We hope the following answers can address your questions and concerns.
>
> A1 (the role of GNN): GNN is adopted to gather variable neighbouring agents' information based on the graph structure. Compared with the fully connected network in MAPPO, it has less computational complexity. We also mentioned in Line 255 that it is implemented with transformer architecture. Combined with the source code in the supplementary material, it should be enough. to reproduce the methods.
>
> A2 (neighbouring group): We constructed such a neighbouring group (agents less than one crossroad), aiming to use as little local information as possible to coordinate agents. Using a more complex neighbouring group is possible, but it will hardly distinguish from the global information in MAPPO, especially with a small number of agents (less than 10) and deviate the insight to use local information.
>
> A3 (incremental development of MAPPO): We would like to emphasize the theoretical improvements over MAPPO: (1) the use of local critic reduces the complexity from $\mathcal{O}(N^2)$ to $\mathcal{O}(GN)$; (2) The corresponding local-critic update method ensures agent coordination even with local information. These factors are important to the successful application in the Flatland environment compared with MAPPO as shown in experiments.
>
> A4 (generalization): The speed and the malfunction $\textbf{DO}$ influence the generalization a lot. During training, all trains have speed with one grid per step and there are no malfunctions. During testing, 3/4 of trains are set slower. It takes more steps for the slow trains to pass the crossroads, and make the passing orders invalid. For the malfunction, there are random malfunctions in the system and the state only includes whether there is a malfunction in the node. Agents have to replan when malfunctions happen for better generalization. The key point here is that there exists large mismatches between training and testing environments and they do influence the planning results. Good learning-based algorithms should not overfit training environments but need to replan based on unseen situations.
>
> A5 (baselines): Please refer to "A1" in the rebuttal to Reviewer 5opN.
>
> We will fix other unreferenced statements and carefully select related sources in the revised version. We hope the reviewer can reconsider the evaluation after clarifying these misunderstandings. Thank you for your time.

---

### Official Review · Reviewer_5opN · 2024-01-22

**Significance And Importance:** 2
**Soundness:** 3
**Novelty:** 2
**Clarity:** 4
**Overall Evaluation:** 1
**Confidence:** 4

**Weaknesses:**

0: Minor weaknesses requiring some work to be addressed for the paper to be accepted.

**Contributions Of The Paper:**

The paper addresses a the MAPF challenge arrises from the FLATLAND challenge, using a multi agent reinforcement learning approach.
The paper leverages CTDE approach and proposes a "local critic" using local group structure to improve scaling and stability.
The proposed LCPPO algorithm outperforms two other standard MARL methods on on FLATLAND scenarios, showing promise for real-world applications.

**Ethical Considerations:**

(1) Not Applicable: The paper does not have any ethical considerations to address

**Nomination For Best Paper:**

No

**Questions For Authors:**

Beyond the weaknesses, just a general concern: what is the relevance to the ICAPS community? unless I missed something, it seems as a pure RL paper, that can can more supportive audience in a learning venue.

**Reproducibility:**

5: Code and domains (whichever apply) are already publicly available

**Strengths Of The Paper:**

The paper addresses an interesting important real world application.  The solution of local critic for groups with the appropriate "tricks" across time is novel.

**Weaknesses Of The Paper:**

comparison to general CTDE methods, without any comparison to RL leader (7th place overall) of the 2020 challenge.

No comparison to the winner the challenge. That case is especially interesting both as it is a planning method and known and relevant to the ICAPS audience,  and second, even if the RL method proposed is inferior to it, the discussion and using it as the bar for RL is important, interesting to see how far RL still needs to go on that application.

It would have been interesting to see some ablation study on the component of LCPPO and not just the final results.

Minor: (de)centralised or (de)centralized - but variation appear in the paper, for consistency choose one.

---

> ### Author Rebuttal · Authors · 2024-01-27
>
> Thank you for your insightful and helpful comments on our paper. We hope the following answers can address your questions and concerns.
>
> A1 (baselines): We agree that the comparison against other OR methods is beneficial to the paper! Thus we compare LCPPO with the winning OR solution of the flatland challenge, thanks to the open-sourced solution: `github.com/Jiaoyang-Li/Flatland`. Notably, we couldn't find detailed implementations of other RL top solutions on this benchmark, so we only baselined standard RL algorithms like IPPO/MAPPO. This makes our work valuable for providing a competitive and easy-to-use RL implementation on this benchmark. We print the average environment reward and time cost for OR and LCPPO methods in the table. The time cost for OR is calculated as the sum of the initial plan and interval replans. The time cost for MARL is calculated as the episode inference time.
>
> | Env Reward / Time Cost (ms) | 10 Agents | 20 Agents  | 30 Agents  |
> | --------------------------- | --------- | ---------- | ---------- |
> | OR                          | 0.93 / 75 | 0.91 / 421 | 0.88 / 862 |
> | LCPPO                       | 0.82 / 75 | 0.71 / 218 | 0.70 / 269 |
>
> Regarding the planning performance, OR consistently outperforms LCPPO, which indicates the big gap left for RL methods. However it is promising to see the time cost for LCPPO doesn't scale as much as the OR method, which results from the CTDE framework. We will add a more detailed comparison in the revised version.
>
> A2 (relevance): We fully understand your concerns. But we do believe that RL as a critical extension of planning methods, is a promising solution to effectively extend the scope and scale of problems (also seen from the time cost in the previous table). In this paper, we employ RL as an approach to solve scheduling problems which is a traditional problem in the field of planning. For this reason, we believe that our paper is relevant to the community of ICCAPS and can attract some attention.
>
> We will add the ablation study on components of LCPPO in the Arxiv version due to the page limit, and fix other minor questions in the revised version. We hope the reviewer can reconsider the evaluation after clarifying these misunderstandings. Thank you for your time.

---

### Meta-Review · Area_Chair_mHFx · 2024-01-31

**Recommendation:** Accept (Poster)
**Confidence:** 4

**Metareview:**

he paper addresses an interesting application of MARL approach to solve a problem which has been well solved by the Operations Research community.

After the rebuttal, the reviewers unanimously agree that there is merit in the paper, though the contribution is not (yet) major, as experimental results show that their MARL approach is still inferior compared with a traditional OR approach.

We like the authors to include comparison with the winning OR solution of the Flatland Challenge in the final version of the paper (which they have included in the rebuttal). They might like also to revise the title of the paper to better reflect its focus.

**Ethical Considerations:**

(1) Not Applicable: The paper does not have any ethical considerations to address